# Spin-polarized two-dimensional electron/hole gases on LiCoO$_2$ layers

**Santosh Kumar Radha and Walter R. L. Lambrecht**

Department of Physics, Case Western Reserve University,
10900 Euclid Avenue, Cleveland, OH-44106-7079

## Abstract

First-principles calculations show the formation of a 2D spin polarized electron (hole) gas on the Li (CoO$_2$) terminated surfaces of finite slabs down to a monolayer, in remarkable contrast with the bulk band structure, which is stabilized by Li donating its electron to the CoO$_2$ layer forming a Co-$d - t_{2g}^6$ insulator. By mapping the first-principles computational results to a minimal tight-binding models corresponding to a non-chiral 3D generalization of the quadripartite Su-Schrieffer-Heeger (SSH4) model and symmetry analysis, we show that these surface states have topological origin.

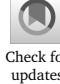

# 1 Introduction

LiCoO$_2$ has been mostly studied as cathode material in Li-ion batteries. [1–3] However, its layered structure also lends itself to the possibility of extracting interesting ultrathin mono- or few layers nanoflakes. A chemical exfoliation procedure has recently been established by Pachuta *et al.* [4] and similar exfoliation studies have also been done on Na$_x$CoO$_2$. [5] The $R\bar{3}m$ structure of LiCoO$_2$ consists of alternating CoO$_2$ layers, which consist of edge sharing CoO$_6$ octahedra, and Li layers stacked in an ABC stacking. By replacing lithium by large organic ions, the distance between the layers swells and they can then be exfoliated in solution and redeposited on a substrate of choice by precipitation with different salts.

Inspired by these experiments we investigated the electronic structure of LiCoO$_2$ few layer systems with various Li and other ion terminations and as function of thickness of the layers using density functional theory (DFT) calculations, of which details are provided in Appendix A. We found, surprisingly that Li no longer fully donates its electron to the CoO$_2$ layer as it does in the bulk but instead a surface state appears above the Li and is occupied with a fraction of an electron per Li forming a two-dimensional electron gas (2DEG). While surface metallization on oxide polar surfaces is often associated with a polar discontinuity and is often avoided by a surface deviation from stoichiometry, [6, 7] this 2DEG formation on Li states is remarkable. The Li bands in bulk LiCoO$_2$ lie at energies $E > 5$ eV above the Fermi level, consistent with the mostly ionic charge donation picture. So, the fact that a Li related surface state comes down sufficiently close to the Fermi level to become partially occupied is truly surprising. Furthermore because it is accompanied by the opposite surface CoO$_2$ becoming spin-polarized it leads actually to a spin-polarized electron gas on the Li side which is located primarily above the Li atoms. The main question we address in this paper is: why does this happen? The answer is that these surface states have a *topological origin*. To be clear, we do not claim that they are topologically protected but that they can be related to a topological surface state of a closely related chiral system. We show that the DFT calculations can be explained by a minimal tight-binding (TB) model, closely related to the quadripartite Su-Schrieffer-Heeger (SSH4) model which has been shown to support topologically protected surface states for specific conditions on the interatomic hopping integrals. [8] We further prove this model-independently in Appendix B by evaluating the inversion operator eigenvalues at time-reversal-invariant momenta (TRIM) in the layer direction. [9] This symmetry indicator is also related to Zak's criterion for Maue-Shockley states. [10] However, while the original SSH4 model has chiral symmetry protecting a metallic surface state at zero energy in the present case, the Li/CoO$_2$ electronegativity difference leads to two energy-separated or gapped surface states which would tend to still place the surface electrons on the (lower energy) CoO$_2$ side. The crucial element that allows the Li surface state to become partially filled is the existence of strong lateral hopping terms between Li atoms on the surface. The resulting band broadening causes the Li surface band to dip below the top of the CoO$_2$ localized surface band leading to a partial electron/hole occupation in these bands respectively. Nonetheless, the reason for the Li bands to come down in energy near the Fermi energy is clearly related to the SSH4 topology.

# 2 Denstity functional results on 2DEG formation

We start by comparing the band structure in bulk LiCoO$_2$ with that of a monolayer LiCoO$_2$ in Fig. 1 (a) and (b). In the bulk case, we find an insulating band structure with a gap between the filled $t_{2g}$ and $e_g$ Co-$d$ bands. This agrees with previous bulk band structure calculations of LiCoO$_2$, e.g [11]. The Li $s$ and $p$ derived bands, highlighted in color occur at high energy indicating that they donate their electron to the Co-$t_{2g}$ orbitals and support a mostly ionic

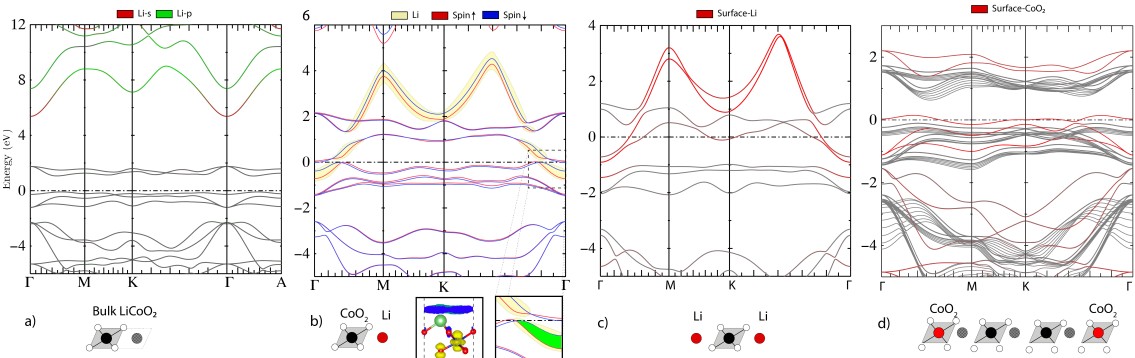

Figure 1: *(a)* Bulk band structure of $LiCoO_2$, *(b)* Spin polarized 2D $LiCoO_2$ monolayer. Insets below, right: zoom in near $\Gamma$ showing spin polarization in green shading, left: electron density of occupied surface state, *(c)* symmetric Li terminated $Li_{x+1}(CoO_2)_x$ with $x = 1$, *(d)* symmetric $CoO_2$ terminated $Li_{x-1}(CoO_2)_x$ with $x = 10$ neglecting spin-polarization. Structural models are shown below each panel.

picture of the bonding. In strong contrast, in the monolayer system, Fig. 1(b), we find an additional set of spin-polarized bands, as highlighted in the figure by yellow shading. Orbital decomposition shows that this band is Li related and its 2D dispersion closely matches that of a hypothetical 2D monolayer of Li atoms. Importantly, it has not only Li-$s$ but also Li-$p_z$ character indicating the formation of Li-$sp_z$ hybrid states. This free-electron-like band has avoided crossings with the Co-$e_g$ bands around 2 eV. It dips down below the Fermi level with an electron pocket near $\Gamma$. Inspection of the corresponding wave function modulo squared shown in the inset below it shows clearly that it is a surface state hovering slightly above the Li atom. This band depends somewhat on the location chosen for the Li atom, (see Appendix C) but its general characteristics are robust. In the lowest energy structure it is found to be spin polarized. This results from the high density of states at the Fermi level due to the rather flat $CoO_2$ surface band which becomes partially filled, which may be viewed as inducing a Stoner instability. For such a partially filled $CoO_2$ layer, correlation effects stemming from the strong on-site Coulomb energies, might be important. Calculations at the DFT+U [12] and quasi-particle self-consistent QS$GW$ levels [13] in Appendix D show that the results are robust with respect to these more accurate treatments of exchange and correlation. The local spin density in the Li-side 2DEG is found to be oppositely polarized from that on the Co-side (see Appendix C). We do not attempt here to study possible alternative orderings of the spins in the plane because of the frustrated nature of the triangular 2D Co lattice. Antiferromagnetic spin polarization of some undetermined ordering direction was found in $Li_xCoO_2$ from the negative Curie-Weiss temperature. [14] The fact that its maximum density lies above the Li atom is consistent with a $sp_z$ hybrid orbital derived band. Bader analysis [15, 16] shows that the surface Li accumulates about $0.4e$ or $\sim 8 \times 10^{14}$ $e/cm^2$. Further calculations for thicker slabs with 3 and 4 layers with the same termination of one Li terminated and one $CoO_2$ terminated surface give very nearly the same surface charge densities. Orbital decomposition in these thicker slabs shows that a second hole doped surface state occurs near the Fermi level on the opposite $CoO_2$ terminated surface layer.

We also consider a symmetrical slab with both surfaces Li terminated. In this case, the system is overcompensated by having one additional Li. The band structure for this case is shown in Fig. 1(c) and has no spin-polarization. A similar surface state is then found on both both Li terminated surfaces and, in fact, one can see that the occupied electron pocket in this band near $\Gamma$ is larger. For symmetry reasons it must contain a fractionalized $e/4$ for each spin and on each surface, so a net charge of $1/2$ electron per Li. While it is perhaps less surprising

to find a metallic state with an odd number of Li atoms, it is still surprising that the Li states come down in energy and host the extra electron rather than donating the electron to the Co and start filling the Co-$d$-$e_g$ band. Similarly, (Fig. 1(d)) symmetric $CoO_2$ termination leads to a surface state on both surface layers with equal hole concentration. This case does prefer a spin-polarized state as shown in Appendix D.

In Appendix C we also considered a 1/2 Li per Co symmetrically at both terminating surfaces. In that case the Li is placed in 1D rows on the surface and a 1D electron gas is found above these Li rows. However, the density of electrons in this case is about 10 times smaller. and the Li surface band, while still present, no longer has sufficient band width to dip below the Fermi level. We further inspect the dilute limit of 1 Li per 4 Co atoms and still find an even smaller residual small charge density in an orbital locally above that Li. This indicates that sufficient lateral hopping between Li atoms is required to generate a significant occupation of the surface states with electrons. Replacing the Li terminating layer by Be (also overcompensating the system from the $CoO_2$ point of view) we find a higher electron density in a Be related surface band. Replacing Li by Na gives similar results but with different band widths of the surface band because of the stronger overlap of the Na orbitals. See Appendix C.

## 3    Tight-binding model and topology

To explain these remarkable results, we now consider a minimal tight-binding model. First, it is clear that the Li needs to be represented by two $sp_z$ orbitals pointing toward the $CoO_2$ layer on either side. It is well known that an even number of orbitals is required in a 1D chiral model to obtain topologically non-trivial band structures. Therefore we choose to represent the $CoO_2$ layer by two $s$-like Wannier orbitals. One could think of these as representing the $a_1$-symmetry of the $D_{3d}$ group or $d_{z^2}$ orbitals on Co with $z$ along the layer stacking **c**-axis making bonding orbitals with O-$p$ on either side of the Co. Of course, this does not represent the full set of $CoO_2$ layer derived bands but we will argue that it represents the relevant bands leading to the surface states. The important point is that the $CoO_2$ and Li each are represented by two Wannier type orbitals whose centers are not on the atoms but on the bonds in between atoms in the layer stacking direction.

This minimal model is a non-chiral version of the SSH4 model. Ordering the orbitals as $\{|Li^a\rangle, |CoO_2^a\rangle, |Li^b\rangle, |CoO_2^b\rangle\}$ the Hamiltonian for the above 1D system (with distance between the layers set to 1) is represented by the following $4 \times 4$ matrix:

$$
\begin{aligned}
H_{1d} &= \begin{pmatrix} \delta & 0 & \tau_1 & \tau_4 e^{ik_z} \\ 0 & -\delta & \tau_2 & \tau_3 \\ \tau_1 & \tau_2 & \delta & 0 \\ \tau_4 e^{-ik_z} & \tau_3 & 0 & -\delta \end{pmatrix} = \begin{pmatrix} \delta\sigma_z & \mathbf{s}^*(k_z) \\ \mathbf{s}(k_z) & \delta\sigma_z \end{pmatrix} \\
&= \sigma_x \otimes \mathbf{h}(k_z) - i\sigma_y \otimes \mathbf{a}(k_z) + \delta\mathbb{1}_2 \otimes \sigma_z,
\end{aligned} \tag{1}
$$

where $\tau_1, \tau_2, \tau_3$, are intra-unit cell hopping parameters while $\tau_4 = \tau_2$ is the out of unit cell hopping parameter, $\delta$ is the ionic on-site term for Li relative to $CoO_2$. The second form of the Hamiltonian focuses on its $2 \times 2$ block structure, in which $\mathbf{s}(k_z)$ is a $2 \times 2$ matrix which is split in its hermitian, $\mathbf{h}(k_z) = \mathbf{h}(k_z)^\dagger$, and anti-hermitian, $\mathbf{a}(k_z)^\dagger = -\mathbf{a}(k_z)$, parts, allowing us to finally write the block structure of the Hamiltonian in terms of the Pauli matrices and a $2 \times 2$ unit matrix $\mathbb{1}_2 = \sigma_0$. For our system, $\tau_2 = \tau_4 = t^z_{Li-CoO_2}$ corresponds to the hopping between the Li and $CoO_2$ layers while $\tau_1 = t^z_{Li} = (E^{Li}_s - E^{Li}_p)/2$ corresponds to the hopping between the two Li $sp_z$'s on the same Li atom, and $\tau_3 = t^z_{CoO_2}$ to O-Co-O hopping within the layer.

This model, which is the SSH4 model for $\delta = 0$ corresponding to chiral symmetry, has been shown [8] to have non-trivial topology which requires zero-energy edge states when

$\tau_1\tau_3 < \tau_2\tau_4$. In fact, in that case, the winding number, which characterizes the topology $\mathcal{W} = \oint \frac{dk_z}{2\pi} \partial_{k_z} \arg \det\{s(k_z)\}$ is 1, while in the other case it is 0. This condition in our case, indicates that the covalent Li-$sp_z$–CoO$_2$ hopping integral is stronger than the intra CoO$_2$ hopping parameter or the Li-$sp_z$ hopping parameter on the same Li atom.

More generally, this non-zero winding number is related to the crystal inversion symmetry and can be deduced from the sign change in the product of occupied eigenvalues $\Pi_n^{occ}\xi_n(k)$ of the inversion operator between eigenstates at $k = 0$ and $k = \pi$, which thus becomes a symmetry indicator [17–19] of the non-trivial topology. This criterion in fact can be checked to be fulfilled for the actual LiCoO$_2$ system in the $z$ direction using the eigenstates of the full DFT Hamiltonian. This model-independent proof of non-trivial topology in the system is detailed in Appendix B and remains relevant even in the non-chiral case. [10]

When $\delta$ is not zero this model becomes non-chiral and the zero energy surface state splits in two, one state moving up and one moving down in energy. Their wavefuncions become localized on opposite edges. The electrons will then localize on the more electronegative side, namely the CoO$_2$ side. Therefore, to explain the electron occupation of the Li-derived surface state, we need to generalize our model to include the lateral in-plane hopping integrals.

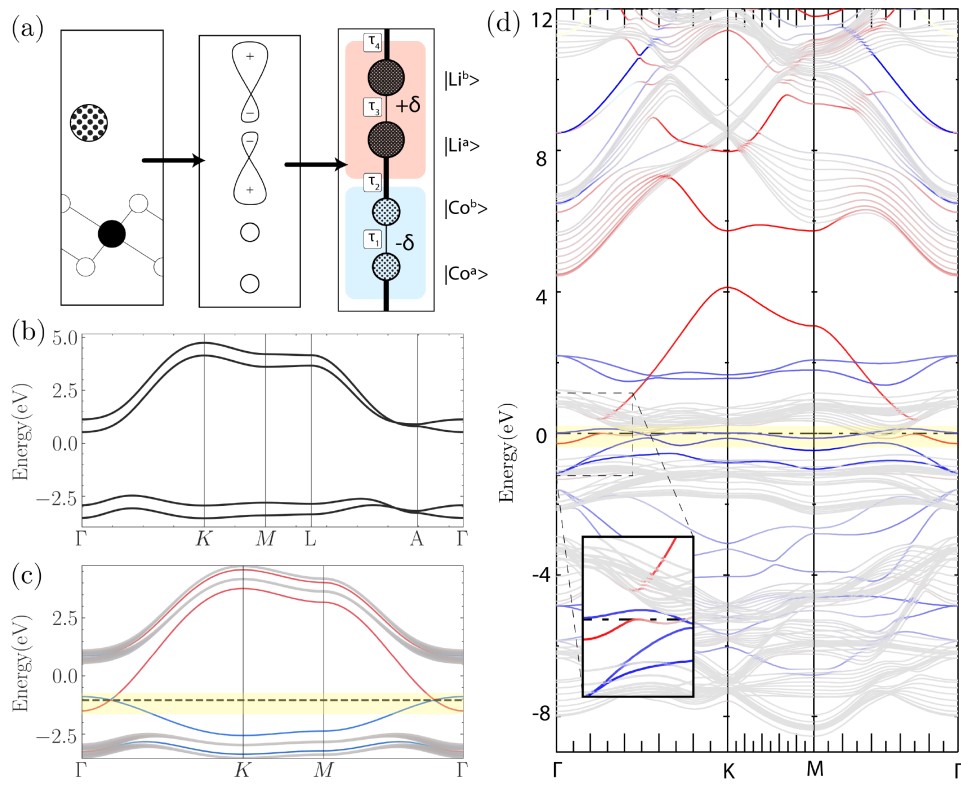

Figure 2: a) LiCoO$_2$ unit cell, 1D-TB model; b) Bulk band structure of 3D Hamiltonian; c) Band structure of finite slab *red* Li surface, *blue* opposite surface CoO$_2$; d) DFT band structure of 14nm slab. (Inset shows surface states near Fermi level.)

We introduce in-plane $t_{\text{Li}}^{xy}$ and $t_{\text{CoO}_2}^{xy}$ hopping paramters on the planar trigonal lattice and define $f_{Li(Co)} = 2t_{Li(Co)}^{xy} \sum_{i=1}^{3} cos(\boldsymbol{k} \cdot \boldsymbol{\delta_i})$ where $\pm\boldsymbol{\delta_i}$ are the 6 vectors pointing toward the nearest neighbors and $\boldsymbol{k}_{\parallel} = k_1\mathbf{b}_1 + k_2\mathbf{b}_2$ is the in-pane 2D wave vector. This modifies only the diagonal terms in (1) leading to

$$H_{3d}(\mathbf{k}_{\parallel}) = \sigma_x \otimes \mathbf{h}(k_z) - i\sigma_y \otimes \mathbf{a}(k_z) + \sigma_0 \otimes [\delta + \Delta(\mathbf{k}_{\parallel})]\sigma_z, \qquad (2)$$

after we drop out a constant, $(\frac{f_{Li}+f_{Co}}{2})$ from the Hamiltonian diagonal. Here, $\Delta(\mathbf{k}_{\parallel}) = \frac{f_{Li}-f_{Co}}{2}$,

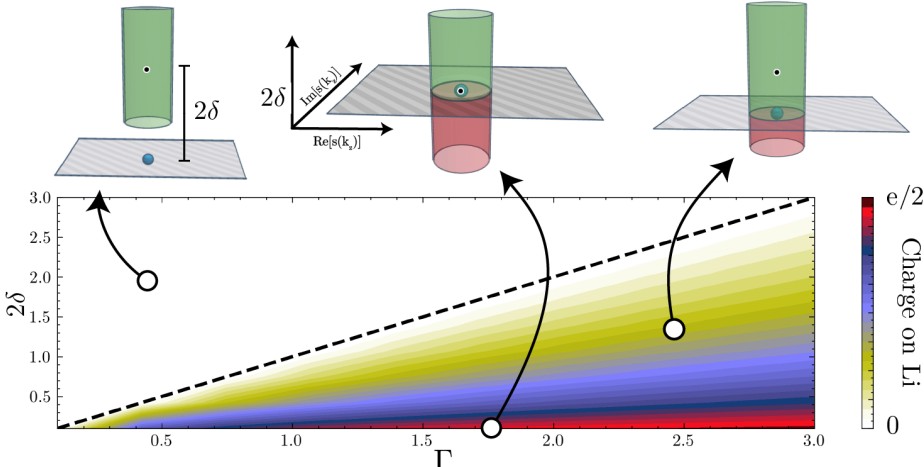

Figure 3: Surface charge density per Li (color map) calculated within the full TB model and cylinder topology.

is added to the $\delta$ in the 1D model and can be thought of as a dimensional crossover parameter, [20] which tunes the influence of the in-plane dimensions. Physically, $\Delta(\mathbf{k}_\parallel)$ is proportional to the width of the energy bands in $\mathbf{k}_\parallel$ space, which is $\Gamma = \Gamma_{Li} + \Gamma_{CoO_2} = 6(|t_{Li}^{xy}| + t_{CoO_2}^{xy})$.

Figure 2(b) shows the band structure of the above $3D$ Hamiltonian while (c) shows the energy levels of the $2D$ periodic system with a finite number of layers along the $z-$axis with top most layer having Li and bottom layer $CoO_2$. While the 3D periodic TB system is seen to have a wide gap between high-lying Li derived bands and low lying $CoO_2$-derived bands, two surface bands with significant $\mathbf{k}_\parallel$ dispersion are seen in part (c), which are respectively localized on the Li (red) and $CoO_2$ (blue) sides and are found to cross each other at the Fermi energy. The interlayer parameters used in the TB Hamiltonian are chosen to satisfy the SSH4 non-triviality criterion and are $t_{Li}^z = 0.5$, $t_{CoO_2}^z = 0.5$ and $t_{Li-CoO_2}^z = 2.0$ eV. The in-plane hopping parameters determining $\Delta(\mathbf{k}_\parallel)$ are chosen to resemble the DFT band structure (shown in part (d) for a 14 nm (30 layers) thick in-plane periodic layer). The opposite dispersion of these surface bands is obvious from the DFT results and translates to these in-plane hopping parameters having opposite sign. In the actual system, it is clear that $t_{Li}^{xy} < 0$ as it is a $\pi$-hopping parameter between Li-$p_z$ states combined with $\sigma$-hopping between the $s$-part of the $sp_z$ orbitals. These hopping integrals involve orbital overlaps with the same sign and a negative potential and are thus negative. In absolute value, $|t_{Li}^{xy}| = 0.6$ is chosen much larger than the $t_{CoO_2}^{xy} = 0.09 > 0$.

The Fermi level is pinned at the intersection of the two surface bands, which indicate an overall semimetallic case with as many holes in the $CoO_2$ surface band as there are electrons in the Li surface band when they overlap. There are just two bands crossing the Fermi level and they cross each other along a nodal line as can be seen in Appendix E. While the crossing is not protected by a different band symmetry label, the red and blue bands are not interacting because they are spatially separated. Obviously in the monolayer an avoided crossing is seen because the smaller spatial distance implies that their interaction is still allowed.

While we recognize that this model is representing only part of the bands of the actual physical system, the correspondence of the surface bands in the DFT and the TB-model Hamiltonian convincingly captures the essence of the relevant physics. Along with the general symmetry indicator argument, we thus conclude that the surface states originate from the topologically nontrivial SSH4 character of the interlayer bonds which correspond to a non-zero winding number. To this is added a term in the Hamiltonian orthogonal to the space of the $\sigma_x$, $\sigma_y$ parts of the Hamiltonian which define a complex plane in which the winding number is defined. One can generally write such a Hamiltonian as $H = \mathbf{d} \cdot \boldsymbol{\sigma}$ where $d_\parallel$ corresponds to

the $(x, y)$ components in a complex plane defining the winding number. [21, 22] The loop defining the winding number is now above or below the complex plane. Its projection on the complex plane encircling the origin (or not) defines the non-zero/zero winding number and therefore topological non-trivial/trivial character. The component $d_\perp$ to this plane determines the energy position of the surface states $E_s = \pm|d_\perp|$, [22] away from zero by the chiral symmetry breaking. In the bipartite SSH model, the third component of the $\boldsymbol{\sigma}$ would be simply the third Pauli matrix $\sigma_z$ while in our case it is $\sigma_0 \otimes \sigma_z$. However, because of the in-plane band dispersion, such an SSH4 like model now applies at each $\mathbf{k}_\parallel$. This turns the loop outside the plane into a cylinder (shown in Fig. 3(b) centered at energy $2\delta$ from the plane with height given by the 2D band width.

The amount of charge on the Li is determined by how much the bottom of the Li band overlaps with the top of the CoO$_2$ band. Assuming a steplike density of states (DOS) for each band near these band edges and parabolic free electron like bands, which makes sense near the band edges for a 2D system, we can easily determine the Fermi energy, located in the surface bands from the fact that the number of electrons in the upper band equals the number of holes in the lower band. Using the density of states for free electrons in 2D to be proportional to the effective mass in each band, and the proportionality of the inverse effective mass to the hopping parameter or band width of the tight-binding model for that band, we find that $q_{\text{Li}}/q_{\text{CoO}_2} = (\Gamma - 2\delta)/(\Gamma + 2\delta)$ if $2\delta > \Gamma$ and zero otherwise. One can easily see that if the splitting of the two surface band centers (which is $2\delta$) is larger than the sum of half their band widths then the charge will still all be localized on the CoO$_2$ and zero on Li. A full numerical calculation of the net surface charge density within the tight-binding model resulting from the overlapping bands is given in Appendix E and shown to closely agree with the above approximate result.

Extending the argument by Mong and Shivamoggi [22], the loop above the plane of the 1D-model becomes extended into a cylinder with a height given by $\Gamma$ and its vertical center placed at a height $2\delta$ above the $\delta = 0$ plane corresponding to the chiral model and hence giving the splitting between the surface states. The part of the cylinder that dips below the surface (shown in red in Fig. 3) gives the amount of charge on the Li side, when assuming a constant DOS along the cylinder in the energy range of band overlap. In Appendix F we show furthermore that the occurrence of the surface states is related to the entanglement spectrum. [23–26]

The above model is consistent with the facts from our DFT calculations that for Na with larger in-plane hopping parameters and hence larger 2D surface band width, a larger surface charge density is found than for Li on the surface. The same is true for Be which also has a smaller electronegativity difference and hence smaller $\delta$ in our model and, in fact gives an additional electron to the surface states. Finally, when two equal surface terminations are used then there is an overall inversion symmetry in the center of the slab and thus requires that the additional electron in the surface states is spread equally over both (symmetry related) sides. The same is true for holes for the case of two CoO$_2$ terminated surfaces.

While we have shown that the presence of surface states in this system have a topological origin, it belongs in the category of topological crystal insulators systems. It does not lead to protection of the metallic surface states because of the breaking of the particle-hole (or chiral) symmetry. This means that certain distortions or surface reconstructions could move the surface states from the gap into the bands. Changes in stoichiometry, such as removing half of the Li atoms on each Li terminated surface is already shown in Appendix C to keep these surface states out of the $t_{2g}$-$e_g$ band gap of LiCoO$_2$ and thereby avoids the 2DEG formation. However, maintaining high enough Li concentration on the surface is a question of controlling the gas environment with which the surface is in equilibrium. While we recognize that the occurrence of the 2DEG is not protected by topology, we maintain that under the correct sur-

face stoichiometry it should be present. An additional concern is whether surface dynamical instabilities could trigger a soft-phonon type instability and thereby avoid the 2DEG formation. This concern is addressed in Appendix G. Other types of surface relaxations, such as changes in the interlayer distance near the surface, which were found to be important in Ref. [27] are already included in our fully relaxed structure DFT calculations.

# 4 Discussion of experimental evidence

While several angular resolved electron spectroscopy (ARPES) and scanning tunneling microscopy studies (STM) have been published in the past [3, 28–32] for both $Li_xCoO_2$ and $Na_xCoO_2$ they were generally focused on the bulk rather than on the search for surface states, which may thus have been missed. As pointed out at various places in our paper, the condition for finding a metallic 2DEG is that the Li concentration on the surface is sufficiently high to allow for the band width of the Li surface band to overlap with the $CoO_2$ surface band to create a semimetallic situation. An X-ray Photoemission Spectroscopy (XPS) study [27] showed that the surface O on the bare $CoO_2$ terminated surface has a reduced negative charge consistent with our prediction of a hole gas on that surface. The ARPES work on $Na_xCoO_2$ on the other hand was mostly focused on $x < 1$ and on the bulk Fermi surface in the context of the superconductivity observed in $Na_xCoO_2$:$yH_2O$ with $x \approx 0.35$, $y = 1.3$. [33] Therefore, the conditions in those studies were not optimal to see the surface state we predict here.

Iwaya *et al.* [3] observed $1 \times 1$ Li islands on their overall $Li_{0.66}CoO_2$ single crystal surface with STM and found a tunneling current consistent with n-type doping in these regions. They interpreted this in terms of an electron counting and purely ionic model in which the Li on the surface (not having a $CoO_2$ layer on top) donates a surplus of electrons to the subsurface $CoO_2$ layer. In other words, each Li gives 1 electron to the one underlying $CoO_2$ layer instead of sharing it equally with two layers. Effectively, they assume it must be the Co transition metal that absorbs the surface change of valence by creating some $Co^{2+}$ rather than the Li. However, this n-type $dI/dV$ signature could also be interpreted as coming from the 2DEG on and above the Li as we propose here. The $dI/dV$ tunneling spectrum in these regions is compatible with a local density of states on the Li-terminated surface as found in our calculations which does show a gap and the Li-related surface band near the bottom of the $e_g$ like conduction band of bulk $LiCoO_2$. Furthermore outside the Li-rich island regions, they found that the bare $CoO_2$ surface regions had metallic character, consistent with our prediction of a hole gas at this surface.

In the ARPES study of Okamoto *et al.* [32], essentially the hole doping effects of reduced Li are observed on the bulk Fermi surface and the results were primarily interpreted in terms of the bulk band structure and the above mentioned surface polar discontinuity arguments. The additional Li-related surface band pocket near $\Gamma$ we predict here was not observed but this may simply be because this surface band is not broad enough to dip below the Fermi level when the Li concentration on the surface is not high enough. While the STM measurements were perhaps able to capture this in specific small regions of $1 \times 1$ full Li coverage, the more global surface measurement of ARPES does not. On the other hand, STM does not allow to probe the band dispersion. Thus we conclude that dedicated surface experiments where the Li concentration on the surface is controlled by additional Li exposure after cleavage may be required to confirm our prediction experimentally. Unfortunately, the Li-surface band we predict is empty over most of the Brillouin zone and thus not observable by photoemission.

# 5 Conclusions

In summary, we have shown that surfaces of LiCoO$_2$ finite slabs host surface states related to the SSH4-like non-trivial interlayer hopping parameters of Li-$sp_z$ and CoO$_2$ bond-centered Wannier orbitals. As a result of strong lateral hopping parameters, the Li related surface band can become partially occupied and host a spin-polarized 2DEG of fairly high electron density. This presents a point of view on the bonding in LiCoO$_2$ different from the prevalent thinking, which traditionally has emphasized the ionic nature of the bonding between Li and CoO$_2$ layers. The fact that this bonding is at least in part covalent and moreover within the non-trivial regime of the hopping parameters and consistent with the inversion symmetry criterion for Shockley surface states has the remarkable consequence of lowering the Li bands at the surface to the point that they can become partially filled. The occurrence of a spin-polarized 2DEG on this surface, which has until now been missed, but could be achieved with adequate surface Li concentration, should be of significant interest to magnetism in 2D systems.

# Acknowledgements

This work was supported by the U.S. Air Force Office of Scientific Research under Grant No. FA9550-18-1-0030. The calculations made use of the High Performance Computing Resource in the Core Facility for Advanced Research Computing at Case Western Reserve University.

# A Computational Methods

Most of the calculations were done using the full-potential linearized muffin-tin-orbital (FP-LMTO) method as implemented in questaal.[34,35] While the main results were derived using density functional theory (DFT) at the generalized gradient approximation GGA in the Perdew-Burke-Ernzerhof (PBE) parametrization [36], we also use the DFT+U approach [37,38] and the quasiparticle self-consistent $GW$ many-body perturbation theory approach.[13,39] Here $G$ and $W$ stand for the one-particle Green's function and screened Coulomb interaction. [40, 41] Convergence parameters for the LMTO calculations were chosen as follows: basis set $spdf - spd$ spherical wave envelope functions plus augmented plane waves with a cut-off of 3 Ry, augmentation cutoff $l_{max} = 4$, $\mathbf{k}$-point mesh, $12 \times 12 \times 2$. The monolayer slabs were separated by a vacuum region of 3 nm. We used experimental lattice constants but relaxed the internal structural parameters.

To determine the parity eigenvalues, as described in Appendix B we used the GPAW code, [42–44] which uses the projector augmented wave (PAW) method [45] and a plane wave type basis set with a cut-off of 600 eV. The results of this method for the band structure of bulk LiCoO$_2$ are in good agreement with those of the FP-LMTO method.

Finally, to study the phonons in monolayer LiCoO$_2$, we used the quantum espresso code. [46] with a plane wave cut-off of 600 eV.

The structural parameters of the systems investigated and python codes for the tight-binding models are available in Ref. [47].

# B Parity eigenvalue

Following the group theoretical analysis of topological crystal insulators the LiCoO$_2$ system is here shown to belong to the category of the obstructed atomic limit. [17, 19, 48] In other

Table 1: Parity eigenvalues at TRIM points in LiCoO$_2$. $n$ is the band index. The number of occupied bands for the hexagonal unit cell($n_{total}$) = 3× (3 Co-d bands + 2× (O-s + 3 O-p) ) = 3 × 11 = 33. One can see that the inversion eigenvalue at $k_z = 0$ and $k_z = \pi$ is inverted for all the TRIM points in the hexagonal Brillouin zone.

| TRIM $k$ | \multicolumn{33}{c|}{Band index} | $\Pi_{n_{occ}}$ |
|---|1|2|3|4|5|6|7|8|9|10|11|12|13|14|15|16|17|18|19|20|21|22|23|24|25|26|27|28|29|30|31|32|33|---|
| $(0,0,0)$ | + | - | + | + | - | - | + | + | - | + | + | - | + | - | + | - | + | - | + | - | + | - | - | - | + | + | - | + | + | - | + | + | - | - |
| $(0,0,\pi)$ | + | - | + | - | + | - | + | - | - | + | - | + | - | - | + | + | - | + | + | - | + | - | + | - | + | - | + | - | + | - | + | + | - | + |
| $(0,\pi,0)$ | - | - | + | + | - | + | + | - | + | - | + | - | - | + | + | - | + | + | - | + | - | - | + | - | + | - | + | - | + | + | + | - | - | - |
| $(0,\pi,\pi)$ | - | + | - | + | + | - | - | + | - | + | - | + | - | - | + | - | + | + | - | + | + | - | + | - | + | - | - | - | + | - | + | - | + | + |
| $(\pi,0,0)$ | - | - | + | + | - | + | + | - | + | - | + | - | - | + | + | - | + | - | + | + | - | + | - | + | - | + | - | + | - | + | + | + | - | - |
| $(\pi,0,\pi)$ | - | + | - | + | + | - | - | + | - | + | - | + | - | - | + | - | + | + | - | + | + | - | + | - | + | - | - | - | + | - | + | - | + | + |
| $(\pi,\pi,0)$ | - | - | + | + | - | + | + | - | + | - | + | - | - | + | + | - | + | + | - | - | + | - | + | - | + | + | - | + | + | + | - | - | - | - |
| $(\pi,\pi,\pi)$ | - | + | - | + | + | - | - | + | - | + | - | + | - | - | - | + | - | + | + | - | + | + | - | + | - | + | - | - | - | + | - | + | - | + |

words, there exists an elementary band representation but it has Wannier centers for the manifold of occupied bands which are not centered on the atomic sites. It is thus "non-trivial" in this sense and expected to have surface states but which are not symmetry protected. This can be ascertained using a symmetry index by examining the eigenvalues of the relevant symmetry operation at Time Reversal Invariant Momenta (TRIM) in the Brillouin zone. In the present case, the relevant symmetry operation is the 3D inversion symmetry operation. Its eigenvalues $\xi_n(\mathbf{k})$ are ±1 and are called parity. As in the case of SSH, the topology of the system is determined by the choice of unit cell. In this case, as we are interested in the layered form of LiCoO$_2$ along the $[0,0,1]$ axis and thus we use the conventional hexagonal cell to describe the rhombohedral $R\bar{3}m$ system. As is well known the **c** axis is then along the direction perpendicular to the layers and the hexagonal conventional cell contains 3 primitive rhombohedral cells in volume corresponding to the rhombohedral $ABC$ stacking. In this hexagonal cell, the TRIM points are the points $(k_1, k_2, k_3)$, where $k_i \in \{0, \pi\}$ after rescaling by the lattice constants along each direction. Using the conventional hexagonal symmetry Brillouin zone, what matters here is comparing the sign of the symmetry indicator at $\Gamma = \{0,0,0\}$ with that at $A = \{0,0,1/2\}$ and $M = \{1/2,0,0\}$ with $L = \{1/2,0,1/2\}$ where $\{\kappa_1, \kappa_2, \kappa_3\}$ give the **k**-point in reduced coordinates, *i.e.* as fraction of $(\mathbf{b}_1, \mathbf{b}_2, \mathbf{b}_3)$.

In Table 1 we show the inversion eigenvalues for different bands and their product $\Pi_n^{occ}\xi_n(\mathbf{k})$. One can see that the inversion eigenvalue at $k_z = 0$ and $k_z = \pi$ are opposite in sign for all the TRIM points in the hexagonal Brillouin zone. This is also shown in Figure 4, which includes all TRIM reciprocal lattice points, including some that are symmetry related in the **c**-plane. This parity eigenvalue reversal in sign leads to non-trivial winding of the wave functions with respect to inversion along the $z$ direction (*i. e.* $[0,0,1]$ axis), just as in the case of previously introduced SSH4 in main text. It is also equivalent to Zak's criterion for having Shockley surface states. [10] In summary, for insulators with inversion, one can distinguish two cases, having in addition chirality or not having chirality. If the system is chiral then the system can still be in a trivial or non-trivial regime depending on the hopping parameters and this is characterized by having a non-zero quantized winding number and can also be ascertained by the above symmetry index. If the system is non-trivial it has protected surface states with a filling anomaly. On the other hand, if it is in the trivial regime no surface states occur. Likewise in the non-chiral case, the system can have hopping parameters in the trivial or non-trivial regime and these two can be distinguished by the non-zero Zak phase or winding number but which now is no longer quantized. But this distinction can still also be made by examining the same symmetry index as before. If this criterion is fulfilled, then there exist surface states but they are no longer protected to remain at fixed energy and they can become gapped, so there is no filling anomaly. It is the latter case our system of LiCoO$_2$ belongs to.

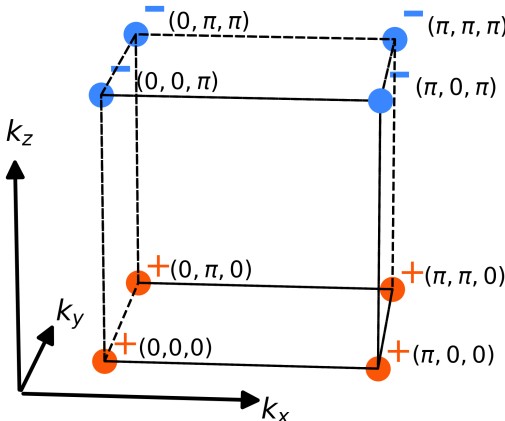

Figure 4: Product of inversion eigenvalue of the occupied band manifold at TRIM points in reduced coordinates. Non trivial inversion winding can be seen along $k_z$ direction with inversion eigenvalues being exchanged.

.

## C    Additional DFT results for various Li surface sites and related systems with Na and Be.

In Fig. 5 we show the changes in band structure for different surface locations of the Li. Their relative total energies are given in Table 2 . The lowest energy position **(1)** for Li is on top of the Co, position **(2)** is above the bottom O, **(4)** above the top layer O and **(3)** in the center of the 2D cell.

Table 2: Relative energy of the structures in Figure 5

| Structure | 1 | 2 | 3 | 4 |
|---|---|---|---|---|
| Energy (eV/f.u.) | 0 | 0.09 | 0.32 | 0.81 |

It is interesting to note that in structure **(3)** shown in Figure 5, the effective hopping between Li and $CoO_2$ layer is mediated by O-$p$ instead of Co-$d$. This change is captured in the band structure by the lowering of the center of the *free-electron* like Li band compared to other cases. On the other hand, this is clearly not the lowest total energy. Interestingly, we find negligible spin-polarization for both locations **(2)** and **(4)**.

At the bottom of Fig. 5 we show the 2D planar averaged electron density and its spin polarization for location **(1)** set against the structure. The region over which we integrated the surface density is to the right of the vertical black line, which is placed just above the Li atom location. This integration gives about 0.25 $e$, while the total Bader charge associated with Li is 0.4 $e$. This shows that 62.5 % of it lies above the Li atom.

In Fig. 6, we show the band structures for various other cases . Part (a) shows the case of a monolayer of $CoO_2$ compensated by one Li per Co but with Li arranged at half the surface density on each surface. The Li atoms then occur in rows. The corresponding electron density

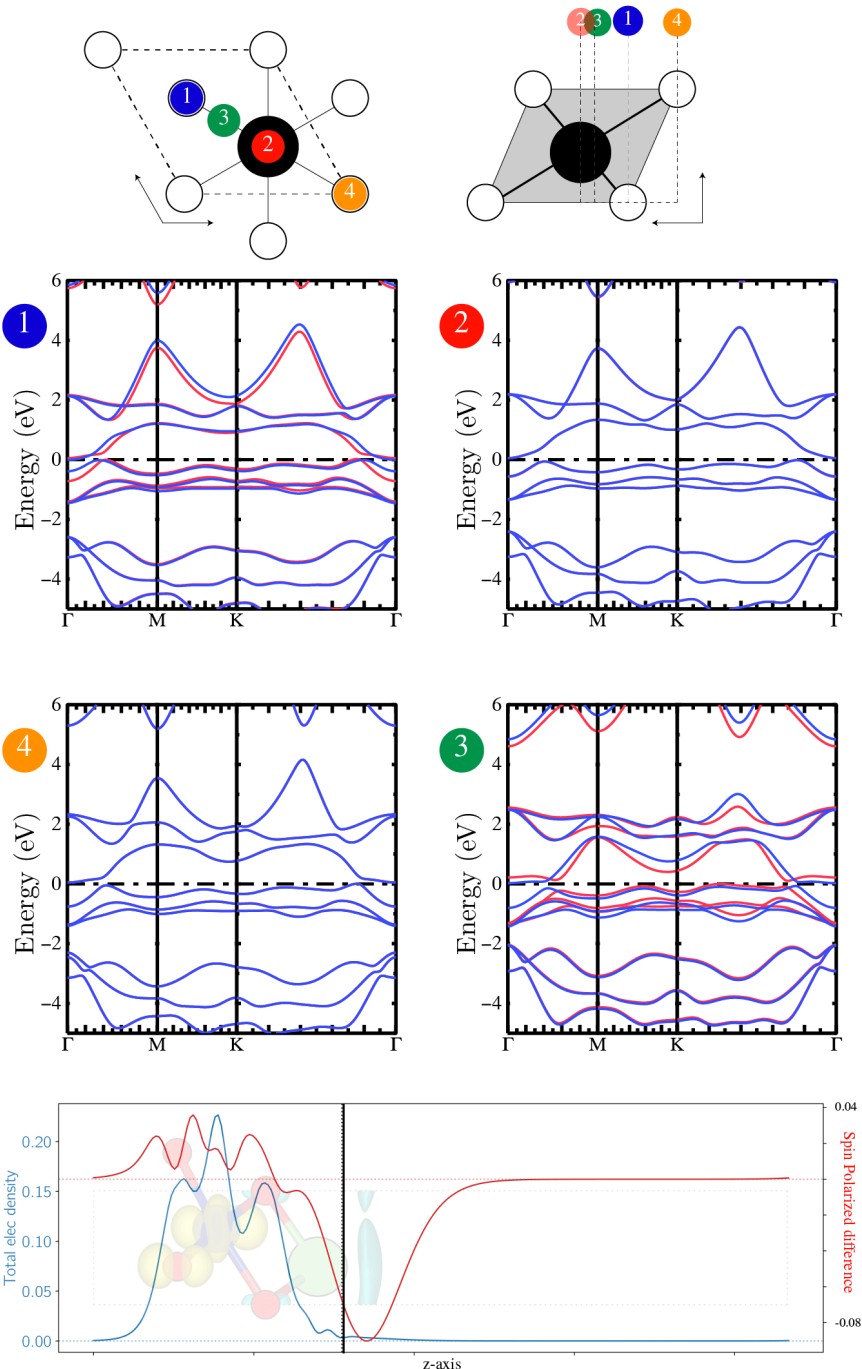

Figure 5: *(top)* top and side view of all 4 symmetric positions of Li on top of 2D CoO$_2$ lattice. *(middle)* Spin polarized GGA band structures for the corresponding structures. *(bottom)* Total electron density (blue) integrated along in-plane axis and spin difference $= n_\downarrow - n_\uparrow$ (red). Black line shows the real space limit for integrating the surface charge density

.

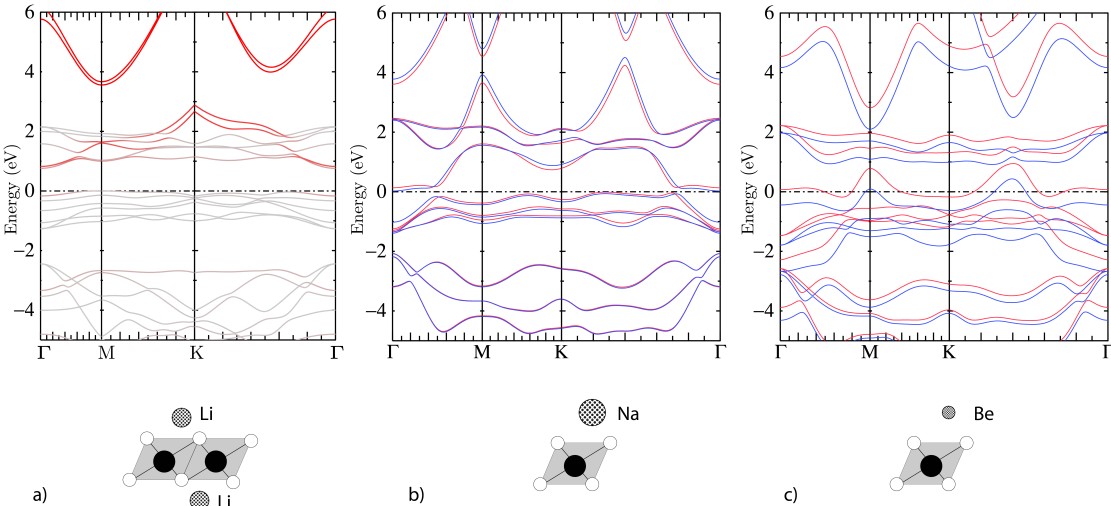

Figure 6: (a) $LiCoO_2$ mono-layer with the Li atoms forming 1D chain (Red color is the Li projected band); (b) $NaCoO_2$; (c) $BeCoO_2$ band structures (*red/blue* bands denote the majority and minority spins).

.

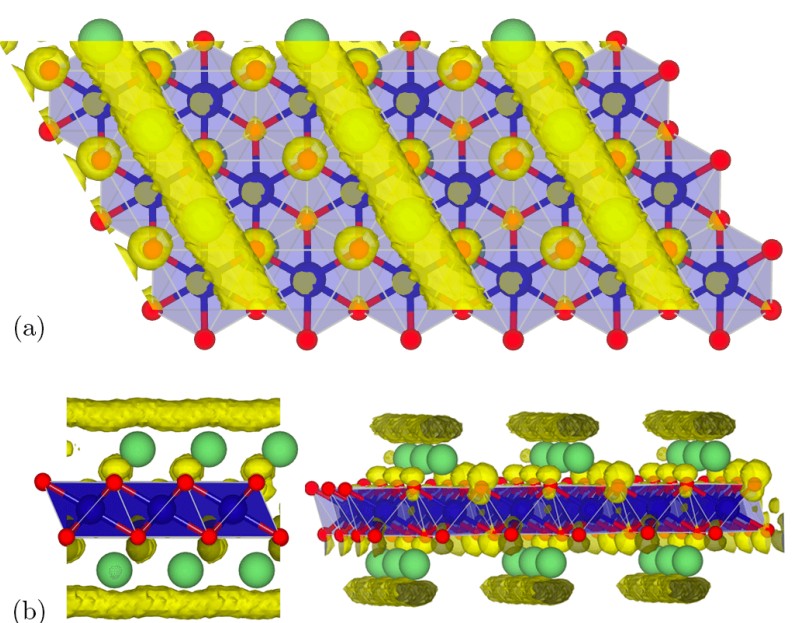

Figure 7: 2D electron density in case of $CoO_2$ monolayer with Li on either side but arranged in rows of Li, achieving one Li per $CoO_2$. yellow isosurface of the electron density correspond to $1.04 \times 10^{-5}$ .

is shown in Fig. 7. While showing some electron density just above the Li atom rows, it should be pointed out that this electron density is a factor 10 times smaller than for the full Li coverage. Correspondingly, we see that the surface bands related to Li do not dip below the Fermi level. Although centered at about the same energy, the band width of this surface band is now about 3 times smaller because the Li only have 2 neighbors (along the 1D rows) instead of 6 in the plane. This prevents this surface band to become occupied. However, from the color coding (red for the Li contribution to the band) we can see that the highest occupied band does contain a slight Li contribution and forms an electron pocket around Γ. The plot of the corresponding wave function modulo squared is what is shown in Fig. 7. This Li row related 1DEG cannot be explained within the SSH4 based tight-binding model. It would require a more complete description of the Li-CoO$_2$ interlayer hoppings.

Next, in Fig. 6(b) we show the case of a fully Na covered CoO$_2$ monolayer with Co on one side. This is similar to the corresponding Li case discussed in the main part of the paper but shows that with Na, the electron pocket around Γ is increased in size. This is consistent with the larger lateral hopping between Na. Finally, in Fig. 6(c) we show the case of Be covered CoO$_2$. Compared to Li, we now overcompensate the CoO$_2$. In this case the electron density in the surface 2DEG is even larger but we also obtain a larger spin-splitting.

# D   Correlation effects and magnetism

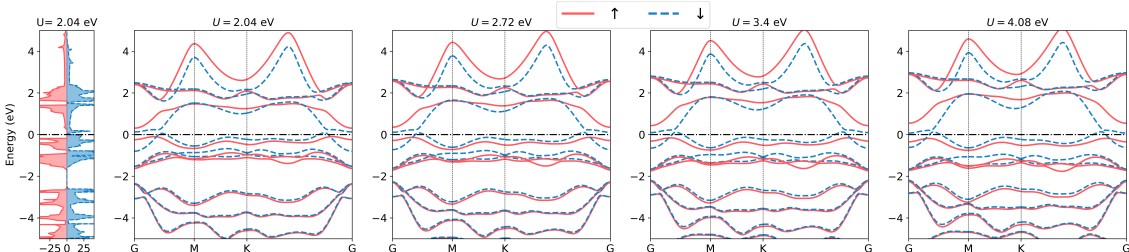

Figure 8: DOS and band structure of majority and minority spins for monolayer LiCoO$_2$ for various values of U

**Local correlations:**   While pure bulk LiCoO$_2$, in which Li fully donates its electron to the CoO$_2$ layers, is clearly a non-magnetic band-insulator based on the electron count with a gap between octahedral filled $t_{2g}$ and empty $e_g$ bands, the incomplete electron donation from the surface Li to surface CoO$_2$ layers leads to a partially filled $d$-$t_{2g}$ band in the monolayer and CoO$_2$ surface regions of the finite slabs. Because of the strong on-site Coulomb interaction on Co-$d$ states, this may lead to correlation effects beyond those treated at the Local Spin Density Approximation (LSDA). While we refer to it here as LSDA, the actual calculations used spin-polarized GGA. In fact, the effect of partial $d$ filling is already manifested in LSDA leading to a spin-polarized solution. A complete treatment of spin correlations due to partial occupancy may require Dynamical Mean Field (DMFT), which is beyond the scope of this study. Here, we investigate whether the qualitative band structure picture in terms of the occurrence of surface states holds up and to what extent the charge accumulated in the Li surface band 2DEG is robust to the inclusion of on-site Coulomb terms within the mean-field type LSDA+U method. To this end Figure 8 shows the DOS and band structure of monolayer LiCoO$_2$ for different values of $U$. We treat the LSDA+U in the simplest form [12] where a potential $V_{m\sigma} = U(\frac{1}{2} - n_{m\sigma})$ is added on orbital $d_{m\sigma}$ as function of its occupation number

$n_{m\sigma}$. This shifts empty states up by $U/2$ and filled states down by $U/2$. Several initial density matrices or occupation numbers are tried out and the results give the one with the lowest total energy. First, one sees no major change in the qualitative picture regarding the non trivial surface bands in the presence of strong on-site Coulomb interaction. Secondly, it can be seen from the DOS that the cause for spin splitting arises from the avoidance of a high density of states at the Fermi level, in other words, the Stoner instability. A DOS peak occurs near the Fermi level because of the almost flat bands near $\Gamma$ (here labeled G). It is important to note that the surface charge concentration is mainly determined by the overlap of the Li and $CoO_2$ surface bands, which in turn depends on the Li surface band width and the on-site potential difference between the effective Li and $CoO_2$ layers. The addition of an effective Hubbard $U$ to the local Co-$d$ states does not disrupt that potential difference (although there might be a slight effect because of Li-Co-O hybridization). This can be seen more quantitatively by calculating the Bader charge [15, 16] associated with the Li atom as function of the $U$ value. The Bader approach consists in partitioning the system into regions bounded by a "zero flux" surface, *i.e.* a 2D surface on which the real space charge density is a minimum in the direction perpendicular to the surface. The total charge inside the Bader surface surrounding each atom is then integrated and gives the Bader charge. While not a unique way of apportioning charge to atoms, it is a precisely defined way.

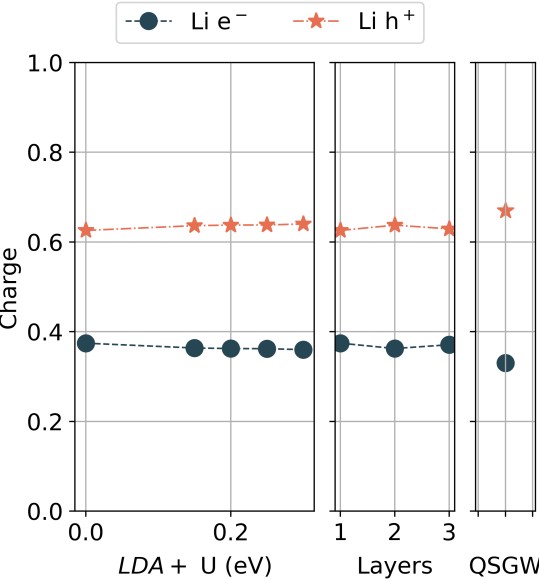

Figure 9: Electron and hole (1-electron) charge on Li atom for (a) various values of U at LDA+$U$ theory (b) LDA for various number of layers (c) QSGW

Figure 9(a) shows the electron and hole (1-electron) Bader charge for various values of $U$ on the Li atom. One can see that there is almost no change in the surface charge content on Li atom as function of $U$. Figure 9(b) also shows the surface charge dependence on the thickness (number of layers) of the finite size sample. Again, because this does not change the on-site potentials on Li vs. Co, no significant change in Li-surface 2DEG charge is observed.

**Non-local correlations:** In LSDA+U only on-site Coulomb terms are added on a particular orbital, the Co-$d$-orbitals and $U$ is essentially treated as an empirical parameter. To account for non-local screening effects, we also study the monolayer band structure using a many-body perturbation theory approach: the quasi-particle self-consistent (QS) $GW$ method, [13, 40] where $G$ and $W$ are the one-electron Green's function and screened Coulomb interaction re-

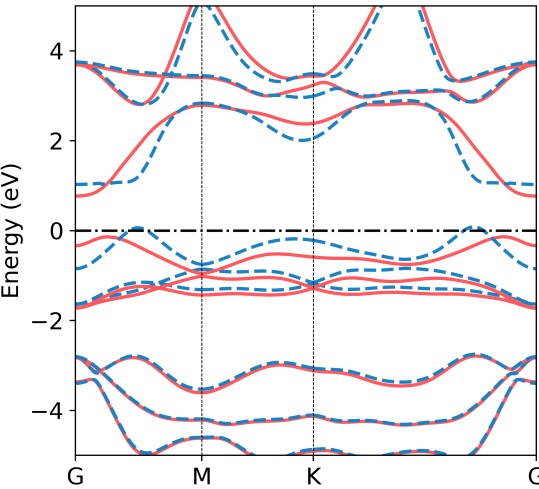

Figure 10: QSGW band structure of the monolayer LiCoO$_2$

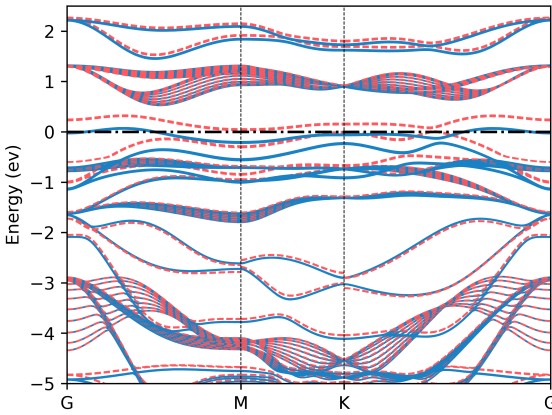

Figure 11: Band structure of Li(CoO$_2$)$_2$ symmetrically terminated slab, including spin-polarization.

spectively. In the QS$GW$ approach, the energy-dependent self-energy $\Sigma(\omega) = iG(\omega) \otimes W(\omega)$ ($\otimes$ means convolution) is replaced by a hermitian energy-averaged non-local exchange-correlation potential used in the independent particle Hamiltonian $H_0$ (defining the $G$ and $W$) and which is iterated to self-consistency. This means that the Kohn-Sham eigenvalues of the independent particle Hamiltonian $H_0$ become the same as the quasi-particle excitation energies of the many-body system and become independent of the starting independent particle Hamiltonian ($H_0$). We here choose the initial $H_0 = H_{LDA}$.

Again, qualitative features of the band structure (shown in Fig. 10) remain intact as the occurrence of surface state is dictated by topology. Because of quasi-particle renormalization due to screening, slight changes in the surface charge is observed as shown in Figure 9(c).

**Magnetic moments** In the main text, we mentioned that for the LiCoO$_2$ monolayer, the spin in the 2DEG on the Li-side is oppositely oriented with respect to that in the Co-layer. That calculation is done within the standard local spin-density approximation (LSDA). We here provide additional information on the magnetic moments in this system. The magnetic moments are summarized in Table 3. The atom contributions are within the muffin-tin sphere

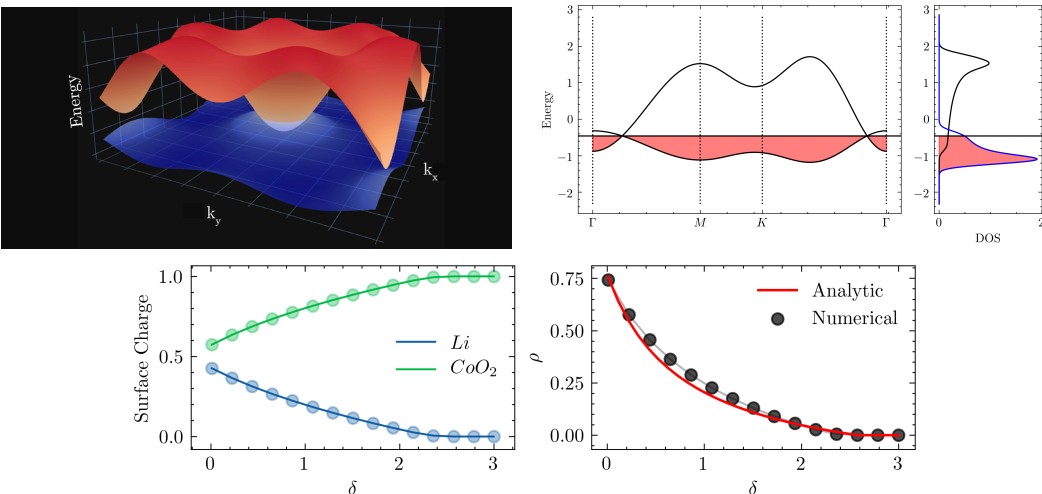

Figure 12: *(Top left)* Intersecting surface bands in TB model, *(Top right)* band structure of surface states and Density of States (DOS) contribution from each band (black on Li side, and blue on $CoO_2$ side) Surface states have a 2D dispersion corresponding to a triangular lattice with nearest neighbor hopping parameters given by $t^{xy}_{CoO_2} = 0.1$ and $t^{xy}_{Li} = -0.3$ in arbitrary units with an on-site offset of $\delta = 0.9$ (in units of $\Gamma/6 = (t^{xy}_{CoO_2} + t^{xy}_{Li})$), *(Bottom left)* Surface charge per unit cell area as function of on-site parameter $\delta$; *(Bottom right)* Ratio of charge on Li to charge on $CoO_2$ surface layers from numerical TB and analytic equation Eq.(6) derived in the main text. In the limit of $\delta = 0$, this ratio approaches 1.

Table 3: Magnetic moment in LiCoO$_2$ monolayer in LSDA+U.

| U (eV) | Co | Li | $O_1$ | $O_2$ | Smooth | Total |
|-------:|------|------|------|-------|--------|-------|
| 0.000 | 0.046 | 0.004 | 0.006 | -0.001 | -0.053 | 0.002 |
| 1.361 | 0.099 | 0.007 | 0.014 | -0.001 | -0.120 | 0.000 |
| 2.041 | 0.092 | 0.007 | 0.015 | -0.001 | -0.114 | 0.000 |
| 2.721 | 0.086 | 0.007 | 0.017 | -0.001 | -0.109 | 0.000 |
| 3.401 | 0.080 | 0.007 | 0.020 | -0.001 | -0.106 | 0.000 |

and the smooth contribution is from the smooth part of the decomposition of the spin density over the whole region and contains the contribution from the interstitial region, in particular the vacuum region above the Li atom (2DEG). $O_1$ and $O_2$ are the surface and non-surface oxygens respectively.

**Spin-polarization for symmetric CoO$_2$ termination:** In Fig. 11 we show the spin-polarized band structure of the symmetrically CoO$_2$ terminated monolayer Li$_9$(CoO$_2$)$_{10}$, which was shown without spin-polarization in the main text in Fig.1d. The origin of spin polarization is again that in this case partially filled Co-$d$ bands occur.

## E Surface charge calculation in TB-model

In Fig. 12 we show the results of a numerical calculation of the surface state occupancy as function of the energy separation of the two surface bands within the tight-binding approximation. We can see that it is in good agreement with the model results in the main paper.

Note that beyond $\delta = 3$ here the overlap of the bands is zero and no charge occurs on the Li side. For $\delta = 0$ we are in the limit where the charge on the Li is $1/2$ by symmetry.

# F    Entanglement spectrum

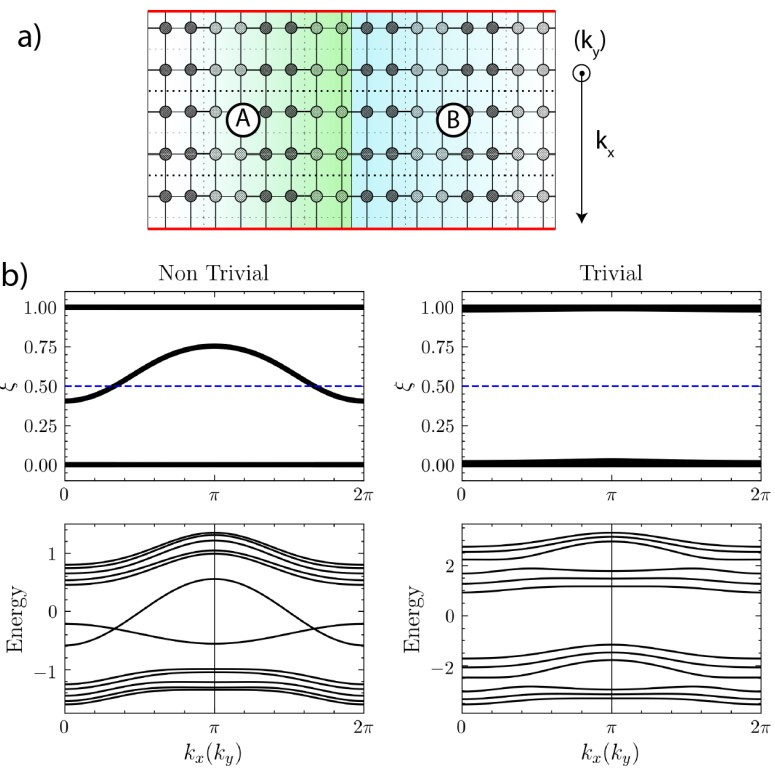

Figure 13: a) Partitioning of the system in two parts for calculating the entanglement spectrum of the reduced 2D system b) *(top)* entanglement spectrum of the 2D system along the cut *(bottom)* band structure of 1D ribbon of the corresponding 2D Hamiltonian showing the edge states for (left) topologically non-trivial and right trivial choice of SSH4 parameters.

To understand the surface states better, we use the idea of *entanglement spectrum* (ES) [49] which has been found to be a generally useful theoretical tool in investigations of topological states [23–26]. The main idea of the ES is that the eigenvalues of the hermitian correlation matrix of the occupied eigenstates, restricted to a subsystem A of the combined system (A+B), provide already information on the existence of surface states when the system would be split in separate A and B parts and of the topologically non-trivial nature of the system. In our case of non-interacting electrons, the correlation matrix is defined in terms of the Bloch functions expanded in the tight-binding basis set as follows. Although our system is periodic in $x$ and $y$ direction we here consider Bloch states only in one direction combined with the layer direction $z$ in which the non-trivial SSH4 topology applies. Let the eigenstates be $|\psi_{k_x}^n\rangle = e^{ik_x x}|u_{nk_x}\rangle = \sum_{j\alpha} e^{ik_x x}[u_{k_x}^n]_{j\alpha}|\phi_{j\alpha}\rangle$, where $i$ labels the sites, which can be either in the A or B part of the system and $\alpha$ labels orbitals per site, then the correlation matrix restricted to the A-subsystems is given by

$$C_{i\alpha,j\beta}^A(k_x) = \sum_n^{occ} [u_{k_x}^n]_{i\alpha}^* [u_{k_x}^n]_{j\beta}, \qquad \text{with } i \in A, j \in A. \tag{3}$$

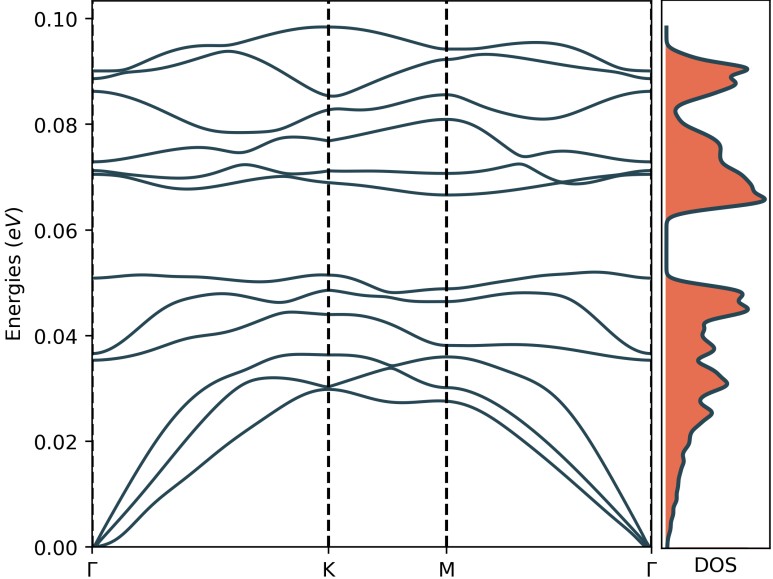

Figure 14: Phonons of monolayer $LiCoO_2$ in the harmonic approximation.

If we remove the restrictions on $i, j$ then we drop the superscript $A$. The eigenvalues of this correlation matrix $\xi(k_x)$ define the ES. If we would not include the restriction, this correlation matrix is built from idempotent projection operators and thus has eigenvalues 0 or 1 only. As shown in [26] and elsewhere, the existence of eigenvalues deviating strongly from 0 or 1, near 1/2 are an indicator of the entanglement of the states between its subparts and thus of the non-trivial topology and the existence of surface states.

The eigenvalues $\xi(k_x)$ of $C_{ij}(k_x)$ matrix at a given $k_x$ are shown in Figure 13(b) along with the 1D eigenvalues of our tight-binding model and clearly show the one-to-one correspondence between the ES containing eigenvalues near 1/2 with the existence of surface states. Furthermore we see that for a different choice of the SSH4 TB parameters, no surface states exist and correspondingly no ES with eigenvalues near 1/2.

The connection between $\xi$ and the previous topological picture in terms of cylinders in Figure 3 of the main paper, is that for each $\mathbf{k}_\parallel$ the eigenvalues $\xi > 0.5$ ($\xi < 0.5$) correspond to a loop above (below) the $z = 0$ plane in Figure 3 of the main paper as indicated by the *green/red* color in the cylinders of the top part of Figure 3. The $\xi = 0.5$ eigenvalue corresponds exactly to the $z = 0$ plane.

# G  Dynamical stability

To address the dynamical stability of the system against soft modes, we have studied the phonons in monolayer $LiCoO_2$ using the Quantum Espresso code along with Phonopy. [46, 50] This is done by constructing a $5 \times 5 \times 1$ supercell and using the phonopy approach. We found that in order to converge these calculations a sufficiently fine **k**-point mesh of $8 \times 8 \times 1$ is required. The resulting phonon dispersion curves are shown in Fig. 14. They show that no soft modes ($\omega^2 < 0$) occur within the harmonic approximation. This ensures that the unreconstructed $1 \times 1$ surface structure is at least dynamically stable.

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
