# Peer review of "Spin-polarized two-dimensional electron/hole gases on LiCoO$_2$ layers."

_SciPost Physics, doi:SciPost Phys. 10, 057 (2021)_

## Round 1 · Referee Report · Anonymous (Referee 1) · 2020-12-17

Strengths

1 - The authors thoroughly analyze their system, using both ab-initio techniques as well as tight-binding descriptions
2 - The manuscript is well organized, starting with the DFT description and following with the discussion based on a tight-binding model.
3 - The number and length of the appendices are appropriate given the material presented in the main text. These appendices provide important details that supplement the main text.

Weaknesses

1 - In some cases, the way in which the authors formulate their results (both in the text and in the figures) could be improved.
2 - It is not always clear how this work relates to the existing theoretical and experimental research on LiCoO2
I will comment in detail on these points in the report below.

Report

Warnings issued while processing user-supplied markup:

  • Inconsistency: plain/Markdown and reStructuredText syntaxes are mixed. Markdown will be used.
    Add "#coerce:reST" or "#coerce:plain" as the first line of your text to force reStructuredText or no markup.
    You may also contact the helpdesk if the formatting is incorrect and you are unable to edit your text.

The authors study surface states appearing in LiCoO2 slabs of finite thickness. They first show that these surface states are present in a DFT simulation of the material. Then they analyze their potential topological origin by using a toy tight-binding model of a multi-orbital SSH chain.

I think the paper is well structured, and I like the combination of ab-initio calculations together with the tight-binding toy models. I believe that exploring the topological origin of surface states in various materials is a timely subject. However, I believe the authors should explain more how their work relates to existing literature, and also improve some aspects of their presentation. These changes would also help to clarify the degree of significance and novelty of their work.

Connection to existing literature:

1) In Phys. Rev. B 82, 075113 (2010), transport properties of LixCoO2 single crystals have shown that the system is insulating close to x=1, in good agreement with the authors bulk DFT band structure. I am confused, however, by the fact that in the thicker slab geometry of the authors' Fig. 2d, the system appears to be metallic. There seem to be gray (bulk) bands crossing the Fermi level. Is this true?

2) The authors should clarify that the DFT band structure of LiCoO2 is not new. It has been done before, for instance in Phys. Rev. B 46, 3729 (1992), which seems to show a good agreement with the authors' results.

3) The STM study done in the authors' Ref. [3] shows that the the CoO2 surface layer is metallic, consistent with the authors' results. However, they find that in some places, the surface is covered by a hexagonal lattice of Li atoms (thus corresponding to the Li termination they discuss). The Li-terminated surface is found to be insulating in experiment, unlike the authors results. This experimental result is further confirmed by the ARPES study of Phys. Rev. B 96, 125147 (2017). Can the authors reconcile their calculation with these results?

4) The authors' Ref. [29] discusses surface relaxation in this material. They find that the spacing between the layers is altered close to the surface, leading to a significant energy difference between the relaxed and un-relaxed slab calculations. Would this impact the surface states?

Improvements to presentation:

These are listed in order of appearance:

1) the insets below the panels in Fig. 1 are too small for me to see, and are also not explained very clearly in the text or figure caption. It's not clear what are the red/blue circles, or why they are cut in half, and why one quarter is hashed out. The spin polarization and electron density mentioned in the caption, are they calculated or just sketched? Does the intensity of the red/blue shading signify the value of a calculated quantity, and if it does, what is the colorscale?

2) on page 2, bottom of left column, the authors say "For symmetry reasons it must contain a fractionalized e/4 for each spin and on each surface." Which symmetry reasons are they referring to?

3) In the discussion of the tight-binding model, the authors sketch the two $sp_z$ orbitals of Li in Fig. 2a. Is this structure of orbitals and their displacement consistent with results from DFT? Are the $sp_z$ orbitals of the bulk system positioned in the same way?

4) On page 3, top of right column, the paragraph starting with "When" contains a very long sentence. It would be helpful to split it into several shorter sentences.

5) What are the tight-binding parameters used in the 1D model of Fig. 2b?

6) The data and sketches of Fig. 3 should be explained in more detail, both in the caption and in the main text. What do the green/red colors represent? How is the charge calculated?

7) On page 4, top of left column, the authors state that $\Delta$ is proportional to $f_{Li} - f_{Co}$, but proportional to $t_{Li}^{xy} + t_{CoO_2}^{xy}$. It would be helpful to clarify the different sign in the two expressions.

8) Below this, they say "it is clear that $t_{Li}^{xy} < 0$ as it is a $\pi$-interaction between between Li-$p_z$ states combined with $\sigma$-interaction between the s-part of the $sp_z$ orbitals." Maybe the authors can clarify this statement more. It is not obvious to me that this should produce a negative hopping.

9) Just below, they say "it means that the Fermi level is pinned at the intersection of the two surface bands." This is also not obvious to me. Maybe it would help to clarify why this Fermi level pinning occurs.

10) On the right column of page 4, the authors say "The topological crystal insulator nature of the system leads to weak topological protection of the surface states." A similar statement is repeated at the top of page 5: "the 2DEG is not strongly protected by topology." I think these statements should be changed. A weak topological insulator is a system which has protected surface states, where (one of) the protecting symmetries is translation symmetry. A strong topological insulator is a system which has protected surface states that do not rely on the presence/absence of translation symmetry. In contrast, the surface states discussed by the authors are not topologically protected in any way. They in fact show that these surface states can be removed from the Fermi level by changing $\Delta$, without breaking any symmetry. Even if these surface states can be related to those appearing in the SSH chain, it should be made clear that this is not a topological system, and there is no topological protection.

11) Finally, I think that the authors should expand the conclusion section to discuss why/how their works fulfills the acceptance criteria of Scipost Physics, as opposed to Scipost Physics Core. As it is written now, the paper seems more suited to the latter.

Requested changes

See report.

  • validity: high
  • significance: good
  • originality: good
  • clarity: good
  • formatting: excellent
  • grammar: excellent

Author:  Santosh Kumar Radha  on 2020-12-25  [id 1109]

(in reply to Report 1 on 2020-12-17)
Category:
answer to question

We thank the referee for his extremely constructive comments.

Comments on connecting to the exisiting literature 1. The fact that bulk Li$_x$CoO$_2$ with $x=1$ is insulating is in agreement with our bulk calculations. The fact that a finite slab, even one of 14 nm or 30 layers, which is still much thinner than a real bulk sample, still shows the metallic surface states in our calculation, shows that what we calculate is a surface effect, not an ultrathin layer effect. In a transport experiment, transport will be dominated by the bulk and special techniques need to be used to identify the surface related transport. So, that the surface metallic states effect has not been identified in transport of bulk samples is not a surprise. The magnetic measurements in PRB82, 075113 (2010) indicate an antiferromagnetic Weiss temperature which could be compatible with our finding of an opposite spin on the CoO$_2$ and Li terminated surfaces, as already mentioned in the manuscript. In the thick slab calculation, the bulk-like bands are indeed depicted in grey and the surface projected bands on the Li side in red. The Fermi level is now pinned by the surface states which occur near the bottom of the conduction band of the bulk. The surface state obviously decays into the bulk and indeed in the region where the grey bands dip below the surface one can see a slight red hue meaning that they are still surface related. However, while the slab is thus technically metallic (because of its metallic surface) the bulk still has a gap. The red and blue lines in Fig. 2d being restricted to the surface layers, should be ignored to see the gapped nature of the bulk of this 30 layer slab. And the slight dip of grey bands in the fermi level are the decaying bluk states that are close to surface, i.e Co bands of the surface LiCoO$_2$ (Note that only the surface Li is colored in red and not the entier surface LiCoO$_2$ layer)

2.Clearly there have been many prior DFT calculations, but usually they did not emphasize the location of the Li derived bands or their $sp_z$ character. The PRB46, 3729 (1992) DFT band structure was done within the atomic spherical wave method. While it agrees with ours in terms of the overall insulating zero spin configuration with occupied $t_{2g}$ and empty $e_g$ bands, it may differ in details like the band gap. In any case the precise band structure of the bulk is not the focus of our paper. However, in the revised verson we include a reference to this paper.

3.Our calculations indicate that the metallic 2DEG on the Li terminated surface can only form if there is sufficient overlap between the Li surface state and the CoO$_2$ surface state. To see this effect experimentally, a precise control over the Li coverage on the surface may be required. The starting point in Iwaya's study is a Li$_x$CoO$_2$ with $x=0.66$ so close to 1/3 of the Li have been removed. However, patches of $1\times1$ Li were found and would indeed correspond to our fully Li covered surface. However, a closer examination of this paper shows that in the regions of $1\times1$ Li coverage they find an n-type tunneling current. Their explanation for this is that with a full local Li coverage, the Li which in bulk donates half an electron to each surrounding CoO$_2$ now donates it to the one CoO$_2$ layer to which it is attached. This leads to a surplus of electrons on the subsurface CoO$_2$ layer, which they measure as n-type. This interpretation makes the plausible but still unproven assumption that Li does not change its valence but the Co in the subsurface CoO$_2$ layer does. This reflects the ingrained thinking about LiCoO$_2$ as having essentially purely ionic bonding between Li and CoO$_2$ and hence a polar discontinuity type of argument is used. However, our paper provides an alternative explanation. The n-type behavior could result directly from the Li-related surface state we predict to exist on the Li termination. While the STM $dI/dV$ does show a gap-like spectrum, this is still compatible with our band structure inf Fig. 2d, which also shows a gap as already explained above in (1). The surface states on the Li surface (in red) occur at the bottom of the bulk conduction bands and are consistent with the STM tunneling observations. The important new insight our paper provides is precisely that one should not think of the bonding in LiCoO$_2$ as being a purely ionic bonding. The directional Li-$sp_z$ mediated bonding plays a decisive role in giving this system an SSH like character, if not for which the conclusions drawn in Iwaya's paper would be correct. We have added a new section in the paper discussing the relation to experiment in more detail.

4.The DFT calculations of both the monolayer (Fig. 2b) and the slab of 30 ,layers (Fig. 2d) were fully structurally relaxed. So, whatever surface interlayer spacing variations occur, they were already included. We did not examine them in detail yet because they are not the focus of our paper.

** Comments on improving the presentation** 1. We have removed the inset part of figure with the circles. They were supposed to convey in a graphical manner how the net charge in the surface states is fixed by symmetry but apparently the figure is not speaking for itself and the explanation is already in the text. 2. It is the inversion symmetry in the center of the slab that connects the top and bottom surfaces in those models which enforces an equal occupation. Note that in this case, the system is also non-stoichimetric and has an additional electron or hole. 3. We have already indicated in the DFT orbital decomposition that $s$ and $p_z$ occur in the high lying bands. On the Li, $s+p_z$ and $s-p_z$ obviously point directly in the $+z$ and $-z$ direction. The other orbitals of our model of course are just hypothetical $s$ orbitals and are not as obviously linked to actual atomic orbitals of $CoO_{2}$. This is after all only a toy model. 4.We appreciate the suggestion and split up the sentence as follows. 'When $\delta$ is not zero this model becomes non-chiral and the zero energy surface state splits in two, one state moving up and one moving down in energy. Their wavefuncions become localized on opposite edges. The electrons will then localize on the more electronegative side, namely the $CoO_2$ side. 5.The TB-parameters were added explicitly in the text. 6. We have added additional explanation about Fig. 3. This cylinder model is introduced to make contact with the topological literature. 7. We thank the referee for pointing out this inconsistency. The point is that the band width should have the $|t^{xy}_{Li}|$ between absolute value signs. It is now corrected and the constant term dropped from the Hamiltonian is also explicitly given. 8. In a $\pi$ interaction the two $p_z$ orbitals have the same sign next to each other ( same orientation of the lobes) but the hopping intgral $\langle p_z|V|p_z\rangle$ involves also a negative potential term. This is well known in tight-binding theory but a sentence is added for the benefit of the reader not familiar with it. 9. Additional explanation is provided about the semimetallic nature and Fermi level position. This can best be seen in Fig. 12 of the appendix. The two surface bands are the only two that are crossing the Fermi level. Furthermore they cross each other because their spatial seperation prevents them from interacting. These two band thus cross along a nodal line. Since we need to exactly half fill these two bands, the Fermi level must occur at this nodal line or in the bands along symmetry lines figure at the linear crossings of the bands. In fact, we can just relabel the bands from bottom to top and the lower one is then filled and upper empty but they touch at the crossing points. 10.We agree with the referee that there is no topological protection of a metallic or partially filled surface state, except in the case of symmetric non-stoichimetric termination of the slab with two identical surfaces. We have changed the text accordingly.

Argument for SciPost Physics

Our argument for acceptance in Scipost Physics as opposed to Scipost Physics Core is that our paper points out two radical new ideas/discovery in a well known system. First, a new point of view on the nature of the bonding in LiCoO$_2$. In contrast to the prevalent thinking that the bonding between Li and CoO$_2$ is purely ionic, our calculations show that the directional $sp_z$ orbital mediated bonding is crucial in giving the system an SSH-like non-trivial topological character, which manifests itself in the existence of a Li related surface state with significantly lower energy near the Fermi level. Second, although not strictly topoligically protected because of the breaking of the particle-hole or chiral symmetry, this surface state has not been identified before and is found here to lead to a spin polarized 2DEG with high carrier density if the Li concentration on the surface is maintained high enough to allow band overlap with the CoO$_2$ surface related surface state. The occurrence of such a 2DEG on this surface is in itself worthy of attention and potentially of interest to devices for people working in 2D electron gas research. Thus the audience for this paper includes both theoritical and experemental physicists/material scientists working in the intersection of applied and fundamental sides.

---

## Round 1 · Referee Report · Anonymous (Referee 2) · 2020-12-18

Strengths

1) The manuscript uses a good combination of density functional theory, and tight-binding microscopic models.

2) Supplements provide other important information that nicely complement the main part of the manuscript, e.g. LSDA+U calculations.

Weaknesses

1) It is not immediately clear how the states reported for Li-terminated Li$_2$(CoO$_2$) can be classified as surface states of a slab.

2) The symmetry analysis with parity eigenvalues is not properly clarified

Report

The authors present density functional theory calculations of LiCoO$_2$ in the bulk, down to the monolayer limit, and considering slabs where they observe surface states. They then proceed to describe the appearance of these surface states with an effective tight-binding model.
My main comments on this manuscript concern the topological analysis the authors provide after they introduce this TB model. As I elaborate below, I believe that the presence of surface states does not derive from any bulk-edge correspondence.

First: surface states can be classified as topological if they are anomalous - meaning that they can be found only on the surface of three-dimensional systems and not in a genuine two-dimensional one.
For instance, the surface states of strong three-dimensional topological insulators are anomalous since they break the fermion doubling theorem: in two-dimension it is impossible to have a system with a single Dirac cone. Similarly, the surface states of topological crystalline insulators are anomalous because they break stronger version of the fermion doubling theorem. It is this anomaly that guarantees the protection of the surface states against generic perturbations and grants them a topological nature. Note that also the surface states of three-dimensional weak topological insulators derive from the stacking of the anomalous helical edge states of quantum spin-Hall insulator. In the present case, however, it is completely unclear to me if and how the surface states carry any anomaly.

Second: topological surface states can appear only on surfaces that are left invariant under a protecting symmetry that can be internal (as time-reversal) or a point group symmetry. In the present case the authors relate the presence of surface state to a "topology" related to the inversion symmetry. In a finite slab there is no surface that is invariant under inversion. Therefore, topological surface states only protected by inversion symmetry should not exist. Only hinge modes protected by inversion symmetry can exist.

These two argument show that the surface states the authors describe are not topological at all. They are Maue-Shockley surface states and the symmetry criterion used by the authors is the one originally devised by Zak (Physical Review B 1985). Note that this criterion is necessary but not sufficient for the presence of the surface states. In other words, the surface states are not topologically required and a proper bulk-boundary correspondence is absent.

Requested changes

1) At the beginning of page 3, the authors state "It is well-known that an even number of orbitals is required in a 1D model to obtain topologically non-trivial band structures.". This statement is certainly true as long as a chiral or particle-hole symmetry is present. But the system described by the authors does not satisfy these internal symmetries. Therefore the authors should not use this argument to justify the choice of their low-energy model.

2) Two paragraphs below they state that the non-zero winding number of a chiral-symmetric model is related to the parity eigenvalues. I do not see in general any connection between the two objects in the model of the authors for the simple reason that the chiral symmetry is broken.

3) In the same paragraph the authors announce a "model-independent proof of non-trivial topology" provided in Appendix B. Here the authors should clearly define what they define by non-trivial topology. Appendix B is essentially the symmetry criterion of Zak for which a proper bulk-edge correspondence does not exist.

4) The subsequent claims of topological non-triviality should be properly redefined keeping in mind the complete absence of protection for the surface states.

5) In page 4 the authors also discuss in detail how the winding number for a chiral insulator is defined. This discussion seems not relevant for the present manuscript since, as also acknowledged by the authors, the chiral symmetry is not present in their model.

6) Towards the end of section III the authors refer to a "weak topological protection". A topological protection has to be defined in terms of certain symmetry breaking perturbation. A weak topological insulator is not protected against translational symmetry breaking perturbations. The surface states of the authors instead do not have any topological protection since even with an inversion symmetric perturbation it might be in principle possible to destroy the surface states.

7) In the conclusions, the claim "topologically required surface states" should be toned down.

8) In Appendix B the authors write "Following the group theoretical analysis of topological crystal insulator in the obstructed atomic limit, the non-trivial topology can often be ascertained". I find this statement very confusing. The theory of topological quantum chemistry (Nature 2017) builds on the theory of elementary band representations. And a topological (crystalline) insulator is defined as a system that cannot be described as the sum of elementary band representation up to the Fermi level. The obstructed atomic insulators the authors mention are not topological according to this scheme.

  • validity: good
  • significance: good
  • originality: good
  • clarity: good
  • formatting: excellent
  • grammar: excellent

Author:  Santosh Kumar Radha  on 2020-12-25  [id 1110]

(in reply to Report 2 on 2020-12-18)
Category:
answer to question

We thank Referee 2 for his comments to which we respond below.

** Comments to the report **

We agree that in the absence of chirality, the surface states we find are not topologically protected in the sense of being fixed at in energy and having a filling anomaly. Nonetheless they are closely related to the corresponding chiral case and it is this relation we wanted to point out. We have added a sentence in the introduction to make this clear and have removed any text that would give the impression of claiming topological protection. We agree with the comments of the referee about the topologically protection in 2D vs. 3D systems but again, we do not claim such topologically protected states to exist in our system. So, these comments essentially are not relevant to the validity of our claims. The same applies to the third paragraph of the referee. We agree that the symmetry criterion we use is equivalent to the one proposed by Zak in the 1985 paper and have now included this reference. However, it can be viewed in retrospect as related to the topological characterization in systems with inversion symmetry. The main point we are trying to make in our paper is that if the corresponding SSH4 chiral model is in the non-trivial limit then breaking the chiral symmetry does not leave the surface states protected but they are still present. They are now gapped and pushed up or down in energy. But if the parent SSH4 system with chiral symmetry is in the trivial regime, then we have no surface states at all. So, in this sense, even though our system is non-chiral, its surface states can still be related to the corresponding chiral counterpart.

** Specific changes requested by the referee **

(1) At the point where we state that an even number of orbitals is required to obtain a non-trivial band structure in 1D, we are still talking about the simpler chiral version from which we build up our model. We've added the word chiral before 1D model to make this clear. (2) The non-zero winding number in the chiral case is related to the parity eigenvalues and at this point we are still talking about the chiral case. The surface states even in the non-chiral case are related to a non-zero winding number of the projection of the loop on the complex plane corresponding to the $\sigma_x$ and $\sigma_y$ parts of the Hamiltonian as explained in Ref. 21, 22. These papers relate the occurrence of Shockley surface states to the topological concepts. (3) By non-trivial topology we mean that the system satisfies the criteria which in the chiral case lead to a topologically protected case and in the non-chiral case lead to the existence of a surface state but which is then not protected. We have rewritten this appendix to clearly define what our definition of non-trivial is and what the different cases are one has to consider in terms of inversion and chirality for a full classification of these systems and the existence and status of topological protection of their surface states. (4) We removed all claims of topological projection. (5) We included the equation for the Zak phase or winding number for the benefit of the reader not versed in the topology. The Zak phase is relevant for both the chiral and non-chiral case but is only quantized for the chiral case. This is now further explained in Appendix B. We refer to Refs. 21, 22 for further discussion of how the Zak phase considerations are generalized for systems without chirality. (6) We changed the text about weak vs. strong topology (which was indeed inaccurate) and removed claims of even weak protection. (7) We removed claims of topologically required surface states and only claim that the surface states is "related" to the unexpected lowering of the Li bands which can be related to the corresponding topological properties in the chiral version of the SSH4 model which provides the basis for our model. These surface states would not exist at all if the system were in a trivial limit in terms of the interlayer interactions in the $z$-direction in terms of the hopping integrals but while we can related their existence to a parity eigenvalue theorem, they are indeed not topologically protected and we now make this unambiguously clear. (8) In the theory of topological quantum chemistry, there are three cases. The trival ionic limit, the strongly topological non-trivial one where no elementary band represention can be ascribed to the bands, and the obstructed atomic insulating case in which such an elementary band representation exists but consists of Wannier orbitals not centered on the atoms. That last case is, in this sense, also "non-trivial" and is the case in which our system belongs. It does not have Dirac cone type surface states which cannot be removed in any way and connect the valence and conduction bands but still has surface states, which occur generally isolated in energy somewhere in the gap but which are not protected in energy location or prevented from splitting. We have now stated this clearly in Appendix B.

---

## Round 1 · Referee Report · Anonymous (Referee 3) · 2020-12-21

Strengths

1) Thorough description and modelling of the DFT results.

Weaknesses

1) The rol of topology in the existence of the surface states is not as clearly shown as claimed.

Report

The authors study different structural models of the material system LiCoO2, including bulk, monolayer and slab geometries. The main result is the existence of a surface 2D electron gas which the authors affirm has a topological origin. The authors base this affirmation on two points. First, the calculation of parity invariants. Second, the mapping to an SSH model.

While I find their DFT results very interesting, the role played by topology in the existence of the surface state is not clear, as explained below. I would only suggest publication of this work if these points are properly addressed.

Requested changes

1) Usually, topologically non-trivial boundary states close the bulk gap. In fact, essentially, they exist because such gap, due to topological reasons, must be closed at the interface with the topologically trivial vacuum that implies the boundary. I fail to see this physics in their results (e.g., which bulk gap is being closed by the surface states). Can the authors elaborate on this?

2) Relation to previous work is unclear. In their introduction, the authors cite Topological quantum chemistry, Nature 547, 298 (2017) when referring to the symmetry indicator that would points to non-trivial topology. However, if I am not mistaken, an inspection of the database derived from this reference, precisely here

https://www.topologicalquantumchemistry.org/#/detail/29225

indicates that this material has been classified as topologically trivial. Can the authors reconcile their results with these results?

Other smaller suggestions regarding presentation:

1) The usage of 'lateral interaction' or similarly trhought the paper to describe hoppings parameters can confuse many readers, since the term 'interaction' regularly implies a proccess involving four fermionic operators. I would suggest to avoid the use the word 'interaction' to describe hoppings.

2) In order to facilitate the reproducibility and the reusability of the author's work it could help to provide the actual lattice parameters and internal coordinates used.

  • validity: ok
  • significance: -
  • originality: -
  • clarity: -
  • formatting: good
  • grammar: -

Author:  Santosh Kumar Radha  on 2020-12-25  [id 1111]

(in reply to Report 3 on 2020-12-21)
Category:
answer to question

Reply to Referee 3 Report

We thank Referee 3 for his opinion that our DFT results are very interesting.

We understand his reservations about the topological origins which were shared by the other referees and in part due to our imperfect wording of our claims. As we also made clear to the other two referees, we do not claim that the system is a topological insulator. Instead we point out that the lowering of the Li states which leads to the surface 2DEG can be related to an SSH-like topological effect in a chiral simplification of the actual system, but which still has some consequences in the actual system.

Response to specific points raised by the referee. ** 1) We agree that in a topological insulator, there are indeed surface states that close the bulk gap. However, there exist a broader family of materials which have some type of non-triviality related to the 1D SSH model which has topologically required edge states under certain conditions on the hopping parameters. The corresponding surface states do not need to span or close the whole gap but can correspond to an isolated band in the gap. This occurs precisely in obstructed atomic limit insulators, which are characterized by occupied band Wannier functions not centered on the atoms but in between on the bonds. The present system of LiCoO$_2$ is shown to fall within this category, which is proven by examining its parity eigenvalues, which predict the existence of Shockley surface states. While these are not topologically protected to occur at a fixed energy or to have a filling anomaly, they still have a topological origin. As we showed with our tight-binding model, the SSH4 chiral model does have a topologically protected edge state. Once chirality is broken it is no longer protected but the surface states still exist. It can now split and become gapped. But in our 3D generalization, the lateral 2D hopping broadens these levels into bands which can still overlap and lead to a semimetallic state. To answer the referee's exact question, the bands that are obstructed are the Li $sp_z$ bands and the band manifolds that belong to CoO$_2$ layer. Thus the gap that gets closed here is the gap between the Li states and CoO$_2$ states. Since the CoO$_2$ states are set of complex hopping terms and interactions, we model it simply by using 2 spatially seperated $s$ states to show this (and the exact result without this approximation is obviously captured in DFT). 2) LiCoO$_2$ in the $R\bar{3}m$ structure is indeed classified as trivial in the topological database. In that context, it means that it can be represented by an elementary band representation. The inversion symmetry criterion we examine in terms of the inversion eigenvalues are most clearly seen in the conventional hexagonal cell rather than the primitive cell as used in this database. That symmetry index given in our Appendix B indicates that the system is in the obstructed atomic limit. This also means it satisfies the Zak criterion for Shockley states. It means that while an elementary band representation exists, which often is interpreted as the system being trivial, it nonetheless does not correspond to a trivial ionic limit in the sense that its occupied bands correspond to Wannier functions which are bond centered. This is precisely also what happens in the SSH and SSH4 1D models under the condition on the hopping parameters which classify these models as being non-trivial. It is in this sense that we say that the origin of the surface states is related to non-trivial topology. 3) We replaced lateral interactions with lateral hopping parameters and throughout the paper where the term interactons was used for hopping. 4) Since we study various systems, bulk, monolayer with different Li locations on the surface, NaCoO$_2$ monolayer, and various thicknesses of slabs, providing all the structural parameters would be too distracting from the main message of the paper. Instead we provide these details, which may indeed be useful for future work, in a more efficient manner on a Zenodo citation to our GitHub project repo at the end of the computational methods section.

---

## Round 2 · Referee Report · Anonymous (Referee 4) · 2021-1-20

Report

The authors have fully addressed all the points I raised during the first round of review. My recommendation is to publish this paper in SciPost Physics as is. As an aside, I want to say that I appreciate very much the fact that they have also published their code and numerical data.

---

## Round 2 · Referee Report · Anonymous (Referee 6) · 2021-2-15

Report

The authors have addressed my questions and now the manuscript more clearly states that the surface states found in this work are not topologically protected. I have no further comments and recommend publication in Scipost Physics.

---

## Round 2 · Referee Report · Anonymous (Referee 5) · 2021-2-15

Report

The authors have addressed my comments. In particular, they have clarified that the surface states they find are not topological per se; they are instead related to the topological surface state of a "parent" chiral symmetric model.
I recommend publication of the revised version of the manuscript in SciPost Physics.

---

## Round 2 · Author Response

Dear Editor,

We have added the requested changes and replies.

Best,
Santosh

---

## Round 2 · List of Changes

Page 1
1) added explecitly the statement that the surface states are not topologically protected
2) reference to Zak's and Maue-Shockley's paper
3) band structure reference

Page 2
1) removed illustration from the figure 1
2)changed interaction to hopping parameters

Page 3
1) changed interaction to hopping parameters

page 4
1)added values of hopping parameters
2) again explained the absence of topological protection of the surface states

page 5
1) added a new section referencing and detailing experemental evidence

page 6/7
1) added more details and discussion on connection to topological states.

other pages
1) changed interaction to hopping parameters

---

## Editorial Decision

published